# RelationNet++: Bridging Visual Representations for Object Detection via Transformer Decoder

**Cheng Chi**[*]
Institute of Automation, CAS
chicheng15@mails.ucas.ac.cn

**Fangyun Wei**
Microsoft Research Asia
fawe@microsoft.com

**Han Hu**
Microsoft Research Asia
hanhu@microsoft.com

## Abstract

Existing object detection frameworks are usually built on a single format of object/part representation, i.e., anchor/proposal rectangle boxes in RetinaNet and Faster R-CNN, center points in FCOS and RepPoints, and corner points in Corner-Net. While these different representations usually drive the frameworks to perform well in different aspects, e.g., better classification or finer localization, it is in general difficult to combine these representations in a single framework to make good use of each strength, due to the heterogeneous or non-grid feature extraction by different representations. This paper presents an attention-based decoder module similar as that in Transformer [31] to bridge other representations into a typical object detector built on a single representation format, in an end-to-end fashion. The other representations act as a set of *key* instances to strengthen the main *query* representation features in the vanilla detectors. Novel techniques are proposed towards efficient computation of the decoder module, including a *key sampling* approach and a *shared location embedding* approach. The proposed module is named *bridging visual representations* (BVR). It can perform in-place and we demonstrate its broad effectiveness in bridging other representations into prevalent object detection frameworks, including RetinaNet, Faster R-CNN, FCOS and ATSS, where about $1.5 \sim 3.0$ AP improvements are achieved. In particular, we improve a state-of-the-art framework with a strong backbone by about $2.0$ AP, reaching $52.7$ AP on COCO test-dev. The resulting network is named RelationNet++. The code is available at https://github.com/microsoft/RelationNet2.

## 1 Introduction

Object detection is a vital problem in computer vision that many visual applications build on. While there have been numerous approaches towards solving this problem, they usually leverage a single visual representation format. For example, most object detection frameworks [9, 8, 24, 18] utilize the rectangle box to represent object hypotheses in all intermediate stages. Recently, there have also been some frameworks adopting points to represent an object hypothesis, e.g., center point in Center-Net [38] and FCOS [29], point set in RepPoints [35, 36, 3] and PSN [34]. In contrast to representing whole objects, some keypoint-based methods, e.g., CornerNet [15], leverage part representations of corner points to compose an object. In general, different representation methods usually steer the detectors to perform well in different aspects. For example, the bounding box representation is better aligned with annotation formats for object detection. The center representation avoids the need for an anchoring design and is usually friendly to small objects. The corner representation is usually more accurate for finer localization.

It is natural to raise a question: *could we combine these representations into a single framework to make good use of each strength?* Noticing that different representations and their feature extractions

---

[*]The work is done when Cheng Chi is an intern at Microsoft Research Asia.

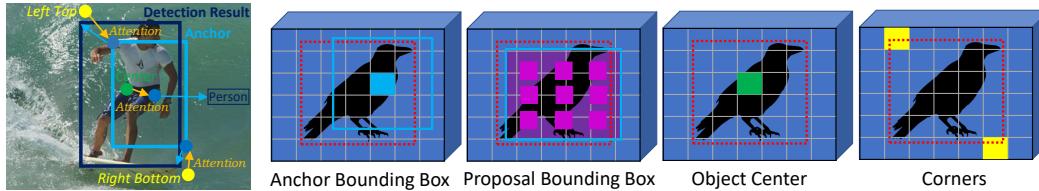

(a) Bridge representations.

Anchor Bounding Box    Proposal Bounding Box    Object Center    Corners

(b) Typical object/part representations.

Figure 1: (a) An illustration of bridging various representations, specifically leveraging corner/center representations to enhance the anchor box features. (b) Object/part representations used in object detection (geometric description and feature extraction). The red dashed box denotes ground-truth.

are usually heterogeneous, combination is difficult. To address this issue, we present an *attention based decoder module* similar as that in Transformer [31], which can effectively model dependency between heterogeneous features. The main representations in an object detector are set as the *query* input, and other visual representations act as the auxiliary *key*s to enhance the *query* features by certain interactions, where both appearance and geometry relationships are considered.

In general, all feature map points can act as corner/center *key* instances, which are usually too many for practical attention computation. In addition, the pairwise geometry term is computation and memory consuming. To address these issues, two *novel* techniques are proposed, including a *key sampling* approach and a *shared location embedding* approach for efficient computation of the geometry term. The proposed module is named *bridging visual representations* (BVR).

Figure 1a illustrates the application of this module to bridge center and corner representations into an anchor-based object detector. The center and corner representations act as *key* instances to enhance the anchor box features, and the enhanced features are then used for category classification and bounding box regression to produce the detection results. The module can work in-place. Compared with the original object detector, the main change is that the input features for classification and regression are replaced by the enhanced features, and thus the strengthened detector largely maintains its convenience in use.

The proposed BVR module is general. It is applied to various prevalent object detection frameworks, including RetinaNet, Faster R-CNN, FCOS and ATSS. Extensive experiments on the COCO dataset [19] show that the BVR module substantially improves these various detectors by $1.5 \sim 3.0$ AP. In particular, we improve a strong ATSS detector by about 2.0 AP with small overhead, reaching 52.7 AP on COCO test-dev. The resulting network is named RelationNet++, which strengthens the relation modeling in [12] from bbox-to-bbox to across heterogeneous object/part representations.

The main contributions of this work are summarized as:

- A general module, named BVR, to bridge various heterogeneous visual representations and combine the strengths of each. The proposed module can be applied in-place and does not break the overall inference process by the main representations.
- Novel techniques to make the proposed bridging module efficient, including a *key sampling* approach and a *shared location embedding* approach.
- Broad effectiveness of the proposed module for four prevalent object detectors: RetinaNet, Faster R-CNN, FCOS and ATSS.

## 2 A Representation View for Object Detection

### 2.1 Object / Part Representations

Object detection aims to find all objects in a scene with their location described by rectangle bounding boxes. To discriminate object bounding boxes from background and to categorize objects, intermediate geometric object/part candidates with associated features are required. We refer to the joint *geometric description* and *feature extraction* as the *representation*, where typical representations used in object detection are illustrated in Figure 1b and summarized below.

**Object bounding box representation** Object detection uses bounding boxes as the final output. Probably because of this, bounding box is now the most prevalent representation. Geometrically, a

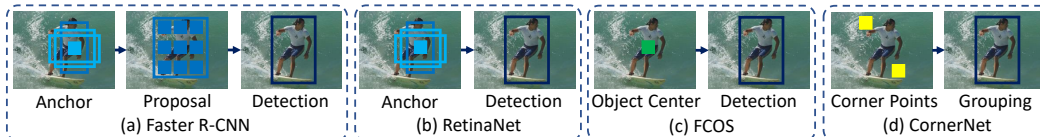

Figure 2: Representation flows for several typical detection frameworks.

bounding box can be described by a 4-d vector, either as center-size $(x_c, y_c, w, h)$ or as opposing corners $(x_{tl}, y_{tl}, x_{br}, y_{br})$. Besides the final output, this representation is also commonly used as initial and intermediate object representations, such as anchors [24, 20, 22, 23, 18] and proposals [9, 4, 17, 11]. For bounding box representations, features are usually extracted by pooling operators within the bounding box area on an image feature map. Common pooling operators include RoIPool [8], RoIAlign [11], and Deformable RoIPool [5, 40]. There are also simplified feature extraction methods, e.g., the box center features are usually employed in the anchor box representation [24, 18].

**Object center representation** The 4-d vector space of a bounding box representation is at a scale of $\mathcal{O}(H^2 \times W^2)$ for an image with resolution $H \times W$, which is too large to fully process. To reduce the representation space, some recent frameworks [29, 35, 38, 14, 32] use the center point as a simplified representation. Geometrically, a center point is described by a 2-d vector $(x_c, y_c)$, in which the hypothesis space is of the scale $\mathcal{O}(H \times W)$, which is much more tractable. For a center point representation, the image feature on the center point is usually employed as the object feature.

**Corner representation** A bounding box can be determined by two points, e.g., a top-left corner and a bottom-right corner. Some approaches [30, 15, 16, 7, 21, 39, 26] first detect these individual points and then compose bounding boxes from them. We refer to these representation methods as *corner representation*. The image feature at the corner location can be employed as the part feature.

**Summary and comparison** Different representation approaches usually have strengths in different aspects. For example, object based representations (bounding box and center) are better in category classification while worse in object localization than part based representations (corners). Object based representations are also more friendly for end-to-end learning because they do not require a post-processing step to compose objects from corners as in part-based representation methods. Comparing different object-based representations, while the bounding box representation enables more sophisticated feature extraction and multiple-stage processing, the center representation is attractive due to the simplified system design.

## 2.2 Object Detection Frameworks in a Representation View

Object detection methods can be seen as evolving intermediate object/part representations until the final bounding box outputs. The representation flows largely shape different object detectors. Several major categorization of object detectors are based on such representation flow, such as *top-down* (object-based representation) vs *bottom-up* (part-based representation), *anchor-based* (bounding box based) vs *anchor-free* (center point based), and *single-stage* (one-time representation flow) vs *multiple-stage* (multiple-time representation flow). Figure 2 shows the representation flows of several typical object detection frameworks, as detailed below.

**Faster R-CNN** [24] employs bounding boxes as its intermediate object representations in all stages. At the beginning, multiple anchor boxes at each feature map position are hypothesized to coarsely cover the 4-d bounding box space in an image, i.e., 3 anchor boxes with different aspect ratios. The image feature vector at the center point is extracted to represent each anchor box, which is then used for foreground/background classification and localization refinement. After anchor box selection and localization refinement, the object representation is evolved to a set of proposal boxes, where the object features are usually extracted by an RoIAlign operator within each box area. The final bounding box outputs are obtained by localization refinement, through a small network on the proposal features.

**RetinaNet** [18] is a one-stage object detector, which also employs bounding boxes as its intermediate representation. Due to its one-stage nature, it usually requires denser anchor hypotheses, i.e., 9 anchor boxes at each feature map position. The final bounding box outputs are also obtained by applying a localization refinement head network.

**FCOS** [29] is also a one-stage object detector but uses object center points as its intermediate object representation. It directly regresses the four sides from the center points to form the final bounding box outputs. There are concurrent works, such as [38, 14, 35]. Although center points can be seen as a degenerated geometric representation from bounding boxes, these center point based methods show competitive or even better performance on benchmarks.

**CornerNet** [15] is built on the intermediate part representation of corners, in contrast to the above frameworks where object representations are employed. The predicted corners (top-left and bottom-right) are grouped according to their embedding similarity, to compose the final bounding box outputs. The detectors based on corner representation usually reveal better object localization than those based on an object-level representation.

## 3 Bridging Visual Representations

For the typical frameworks in Section 2.2, mainly one kind of representation approach is employed. While they have strengths in some aspects, they may also fall short in other ways. However, it is in general difficult to combine them in a single framework, due to the heterogeneous or non-grid feature extraction by different representations. In this section, we will first present a general method to bridge different representations. Then we demonstrate its applications to various frameworks, including RetinaNet [18], Faster R-CNN [24], FCOS [29] and ATSS [37].

### 3.1 A General Attention Module to Bridge Visual Representations

Without loss of generality, for an object detector, the representation it leverages is referred to as the *master* representation, and the general module aims to bridge other representations to enhance this *master* representation. Such other representations are referred to as *auxiliary* ones.

Inspired by the success of the decoder module for neural machine translation where an attention block is employed to bridge information between different languages, e.g., Transformer [31], we adapt this mechanism to bridge different visual representations. Specifically, the *master* representation acts as the *query* input, and the *auxiliary* representations act as the *key* input. The attention module outputs strengthened features for the *master* representation (*queries*), which have bridged the information from *auxiliary* representations (*keys*). We use a general attention formulation as:

$$\mathbf{f}_i'^q = \mathbf{f}_i^q + \sum_j S\left(\mathbf{f}_i^q, \mathbf{f}_j^k, \mathbf{g}_i^q, \mathbf{g}_j^k\right) \cdot T_v(\mathbf{f}_j^k), \tag{1}$$

where $\mathbf{f}_i^q, \mathbf{f}_i'^q, \mathbf{g}_i^q$ are the input feature, output feature, and geometric vector for a *query* instance $i$; $\mathbf{f}_j^k, \mathbf{g}_j^k$ are the input feature and geometric vector for a *key* instance $j$; $T_v(\cdot)$ is a linear *value* transformation function; $S(\cdot)$ is a similarity function between $i$ and $j$, instantiated as [12]:

$$S\left(\mathbf{f}_i^q, \mathbf{f}_j^k, \mathbf{g}_i^q, \mathbf{g}_j^k\right) = \text{softmax}_j\left(S^A(\mathbf{f}_i^q, \mathbf{f}_j^k) + S^G(\mathbf{g}_i^q, \mathbf{g}_j^k)\right), \tag{2}$$

where $S^A(\mathbf{f}_i^q, \mathbf{f}_j^k)$ denotes the appearance similarity computed by a scaled dot product between *query* and *key* features [31, 12], and $S^G(\mathbf{g}_i^q, \mathbf{g}_j^k)$ denotes a geometric term computed by applying a small network on the relative locations between $i$ and $j$, i.e., cosine/sine location embedding [31, 12] plus a 2-layer MLP. In the case of different dimensionality between the *query* geometric vector and *key* geometric vector (4-d bounding box vs. 2-d point), we first extract a 2-d point from the bounding box, i.e., center or corner, such that the two representations are homogeneous in geometry description for subtraction operations. The same as in [31, 12], multi-head attention is employed, which performs substantially better than using single-head attention. We use an attention head number of 8 by default.

The above module is named *bridging visual representations* (BVR), which takes *query* and *key* representations of any dimension as input and generates strengthened features for the *query* considering both their appearance and geometric relationships. The module can be easily plugged into prevalent detectors as described in Section 3.2 and 3.3.

### 3.2 BVR for RetinaNet

We take RetinaNet as an example to showcase how we apply the BVR module to an existing object detector. As mentioned in Section 2.2, RetinaNet adopts anchor bounding boxes as its *master* representation, where 9 bounding boxes are anchored at each feature map location. Totally, there are $9 \times H \times W$ bounding box instances for a feature map of $H \times W$ resolution. BVR takes the

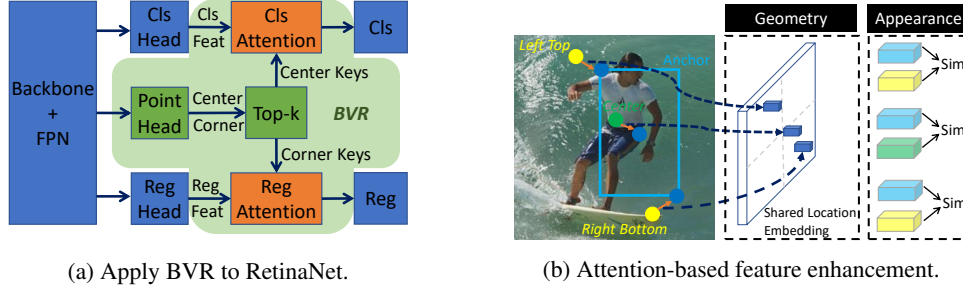

(a) Apply BVR to RetinaNet.

(b) Attention-based feature enhancement.

Figure 3: Applying BVR into an object detector and an illustration of the attention computation.

$C \times 9 \times H \times W$ feature map ($C$ is the feature map channel) as *query* input, and generates strengthened *query* features of the same dimension.

We use two kinds of *key* (*auxiliary*) representations to strengthen the *query* (*master*) features. One is the object center, and the other is the corners. As shown in Figure 3a, the center/corner points are predicted by applying a small point head network on the output feature map of the backbone. Then a small set of *key* points are selected from all predictions, and are fed into the attention modules to strengthen the classification and regression feature, respectively. In the following, we provide details of these modules and the crucial designs.

**Auxiliary (key) representation learning** The point head network consists of two shared $3 \times 3$ conv layers, followed by two independent sub-networks (a $3 \times 3$ conv layer and a sigmoid layer) to predict the scores and sub-pixel offsets for center and corner prediction, respectively [15]. The score indicates the probability of a center/corner point locating at the feature map bin. The sub-pixel offset $\Delta x, \Delta y$ denotes the displacement between its precise location and the top-left (integer coordinate) of each feature bin, which accounts for the resolution loss by down-sampling of feature maps.

In learning, for the object detection frameworks with an FPN structure, we assign all ground-truth center/corner points to all feature levels. We find it performs better than the common practice where objects are assigned to a particular level [17, 18, 29, 15, 35], probably because it speeds up the learning of center/corner representations due to more positive samples employed in each level. The focal loss [18] and smooth L1 loss are employed for the center/corner score and sub-pixel offset learning, with loss weights of $0.05$ and $0.2$, respectively.

**Key selection** We use corner points to demonstrate the processing of auxiliary representation selection since the principle is same for center point representation. We treat each feature map position as an object corner candidate. If all candidates are employed in the *key* set, the computation cost of BVR module is unaffordable. In addition, too many background candidates may suppress real corners. To address these issues, we propose a top-$k$ ($k = 50$ by default) *key* selection strategy. Concretely, a $3 \times 3$ MaxPool operator with stride 1 is performed on the corner score map, and the top-$k$ corner candidates are selected according to their corner-ness scores. For an FPN backbone, we select the top-$k$ *key*s from all pyramidal levels, and the *key* set is shared by all levels. This *shared key* set outperforms that of independent *key* set for different levels, as shown in Table 1.

**Shared relative location embedding** The computation and memory complexities for direct computation of the geometry term are $\mathcal{O}(\text{time}) = (d_0 + d_0 d_1 + d_1 G) K H W$ and $\mathcal{O}(\text{memory}) = (2 + d_0 + d_1 + G) K H W$, respectively, where $d_0, d_1, G, K$ are the cosine/sine embedding dimension, inner dimension of the MLP network, head number of the multi-head attention module and the number of selected *key* instances, respectively. As shown in Table 3, the default setting ($d_0 = 512, d_1 = 512, G = 8, K = 50$) is time-consuming and space-consuming.

Noting that the range of relative locations are limited, i.e., $[-H + 1, H - 1] \times [-W + 1, W - 1]$, we apply the cosine/sine embedding and the 2-layer MLP network on a fixed 2-d relative location map to produce a $G$-channel geometric map, and then compute the geometric terms for a *key/query* pair by bilinear sampling on this geometric map. To further reduce the computation, we use a 2-d relative location map with the unit length $U$ larger than 1, e.g., $U = 4$, where each location bin indicates a length of $U$ in the original image. In our implementation, we adopt $U = \frac{1}{2}S$ ($S$ indicates the stride of the pyramid level) and a location map of $400 \times 400$ resolution, which accounts for a $[-100S, 100S) \times [-100S, 100S)$ range on the original image for a pyramid level of stride S. Figure 3b gives an illustration. The computation and memory complexities are reduced to

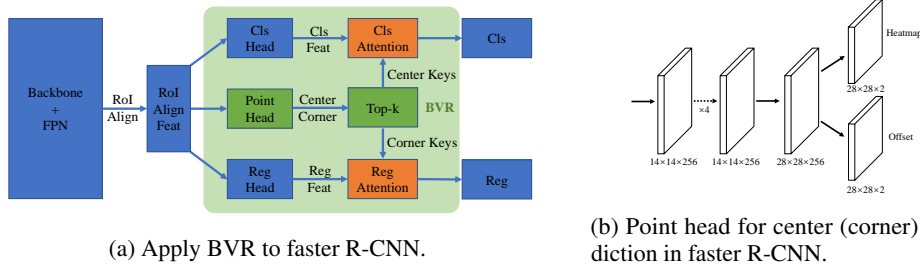

(a) Apply BVR to faster R-CNN.

(b) Point head for center (corner) prediction in faster R-CNN.

Figure 4: Design of applying BVR to faster R-CNN.

$\mathcal{O}(\text{time}) = (d_0 + d_0 d_1 + d_1 G) \cdot 400^2 + GKHW$ and $\mathcal{O}(\text{memory}) = (2 + d_0 + d_1 + G) \cdot 400^2 + GKHW$, respectively, which are significantly smaller than direct computation, as shown in Table 3.

**Separate BVR modules for classification and regression** Object center representations can provide rich context for object categorization, while the corner representations can facilitate localization. Therefore, we apply separate BVR modules to enhance classification and regression features respectively, as shown in Figure 3a. Such separate design is beneficial, as demonstrated in Table 5.

### 3.3 BVR for Other Frameworks

The BVR module is general, and can be applied to other object detection frameworks.

**ATSS** [37] applies several techniques from anchor-free detectors to improve the anchor-based detectors, e.g. RetinaNet. The BVR used for RetinaNet can be directly applied.

**FCOS** [29] is an anchor-free detector which utilizes center point as object representation. Since there is no corner information in this representation, we always use the center point location and the corresponding feature to represent the *query* instance in our BVR module. Other settings are maintained the same as those for RetinaNet.

**Faster R-CNN** [24] is a two-stage detector which employs bounding boxes as the inter-mediate object representations in all stages. We adopt BVR to enhance the features of bounding box proposals, the diagram is shown in Figure 4a. In each of the proposals, RoIAlign feature is used to predict center and corner representations. Figure 4b shows the network structure of point (center/corner) head, which is similar with mask head in mask R-CNN [11]. The selection of *key*s is same with the process in RetinaNet, which is stated in Section 3.2. We use the features interpolated from the point head as the key features, center and corner features are also employed to enhance classification and regression, respectively. Since the number of the *querys* is much smaller than that in RetinaNet, we directly compute the geometry term other than using the shared geometric map.

### 3.4 Relation to Other Attention Mechanisms in Object Detection

**Non-Local Networks (NL) [33] and RelationNet [12]** are two pioneer works utilizing attention modules to enhance detection performance. However, they are both designed to enhance instances of a single representation format: non-local networks [33] use self-attention to enhance a pixel feature by fusing in other pixels' features; RelationNet [12] enhance a bounding box feature by fusing in other bounding box features.

In contrast, BVR aims to bridge representations in different forms to combine the strengths of each. In addition to this conceptual difference, there are also new techniques in the modeling aspect. For example, techniques are proposed to enable homogeneous difference/similarity computation between heterogeneous representations, i.e., 4-d bounding box vs 2-d corner/center points. Also, there are new techniques proposed to effectively model relationship between different kinds of representations as well as to speed-up computation, such as *key* representation selection, and the shared relative location embedding approach. The proposed BVR is actually complementary to these pioneer works, as shown in Table 7 and 8.

**Learning Region Features (LRF) [10] and DeTr [1]** use an attention module to compute the features of object proposals [10] or querys [1] from image features. BVR shares similar formulation as them, but has a different aim to bridge different forms of object representations.

Table 1: Ablation on key selection approaches

| #keys | share | AP | $AP_{50}$ | $AP_{75}$ |
|---|---|---|---|---|
| - | - | 35.6 | 55.5 | 39.0 |
| 20 | ✗ | 36.1 | 54.9 | 39.6 |
| 50 | ✗ | 37.0 | 55.8 | 40.6 |
| 20 | ✓ | 37.7 | 56.5 | 41.4 |
| 50 | ✓ | **38.5** | **57.0** | **42.3** |
| 100 | ✓ | 38.3 | 56.9 | 42.0 |
| 200 | ✓ | 38.2 | 56.7 | 41.9 |

Table 2: Ablation of sub-pixel corner/centers

| CLS (ct.) | REG (cn.) | AP | $AP_{50}$ | $AP_{75}$ | $AP_{90}$ |
|---|---|---|---|---|---|
| - | - | 35.6 | 55.5 | 39.0 | 9.3 |
| integer | integer | 37.0 | 55.6 | 40.8 | 11.0 |
| integer | sub-pixel | 38.0 | 56.1 | 41.7 | 12.5 |
| sub-pixel | integer | 37.2 | 56.7 | 41.2 | 10.4 |
| sub-pixel | sub-pixel | **38.5** | **57.0** | **42.3** | **12.6** |

Table 3: Effect of shared relative location embedding

| geometry | memory | FLOPs | AP | $AP_{50}$ | $AP_{75}$ |
|---|---|---|---|---|---|
| baseline | 2933M | 239G | 35.6 | 55.5 | 39.0 |
| appearance only | 3345M | 264G | 37.4 | 56.7 | 40.4 |
| non-shared | 9035M (+5690M) | 468G (+204G) | 38.3 | **57.2** | 41.7 |
| shared | 3479M (+134M) | 266G (+2G) | **38.5** | 57.0 | **42.3** |

Table 4: Comparison of different unit length and size of the shared location map

| unit length | size | AP | $AP_{50}$ | $AP_{75}$ |
|---|---|---|---|---|
| $[2, 4, 8, 16, 32]$ | $400 \times 400$ | 38.2 | 56.7 | 41.8 |
| $[4, 8, 16, 32, 64]$ | $400 \times 400$ | **38.5** | **57.0** | **42.3** |
| $[8, 16, 32, 64, 128]$ | $400 \times 400$ | 38.4 | 56.8 | 42.2 |
| $[4, 8, 16, 32, 64]$ | $800 \times 800$ | 38.3 | 56.9 | 42.1 |
| $[4, 8, 16, 32, 64]$ | $200 \times 200$ | 38.1 | 56.7 | 41.8 |

# 4 Experiments

We first ablate each component of the proposed BVR module using a RetinaNet base detector in Section 4.1. Then we show benefits of BVR applied to four representative detectors, including two-stage (i.e., faster R-CNN), one-stage (i.e., RetinaNet and ATSS) and anchor-free (i.e., FCOS) detectors. Finally, we compare our approach with the state-of-the-art methods.

Our experiments are all implemented on the MMDetection v1.1.0 codebase [2]. All experiments are performed on MS COCO dataset[19]. A union of $80k$ train images and a $35k$ subset of val images are used for training. Most ablation experiments are studied on a subset of $5k$ unused val images (denoted as `minival`). Unless otherwise stated, all the training and inference details keep the same as the default settings in MMDetection, i.e., initializing the backbone using the ImageNet [25] pretrained model, resizing the input images to keep their shorter side being $800$ and their longer side less than or equal to $1333$, optimizing the whole network via the SGD algorithm with $0.9$ momentum, $0.0001$ weight decay, setting the initial learning rate as $0.02$ with the $0.1$ decrease at epoch $8$ and $11$. In the large model experiments in Table 10 and 12, we train 20 epochs and decrease the learning rate at epoch 16 and 19. Multi-scale training is also adopted in large model experiments, for each mini-batch, the shorter side is randomly selected from a range of $[400, 1200]$.

## 4.1 Method Analysis using RetinaNet

Our ablation study is built on a RetinaNet detector using ResNet-50, which achieves 35.6 AP on COCO `minival` ($1\times$ settings). Components in the BVR module are ablated using this base detector.

**Key selection** As shown in Table 1, compared with independent keys across feature levels, sharing keys can bring $+1.6$ and $+1.5$ AP gains for 20 and 50 keys, respectively. Using 50 keys achieves the best accuracy, probably because that too few keys cannot sufficiently cover the representative keypoints, while too large number of keys include many low-quality candidates.

On the whole, the BVR enhanced RetinaNet significantly outperforms the original RetinaNet by 2.9 AP, demonstrating the great benefit of bridging other representations.

**Sub-pixel corner/center** Table 2 shows the benefits of using sub-pixel representations for centers and corners. While sub-pixel representation benefits both classification and regression, it is more critical for the localization task.

**Shared relative location embedding** As shown in Table 3, compared with direct computation of position embedding [12], the proposed shared location embedding approach saves $42\times$ memory cost (+134M vs +5690M) and saves $102\times$ FLOPs (+2G vs +204G) in the geometry term computation, while achieves slightly better performance (38.5 AP vs 38.3 AP).

Table 5: Effect of different representations ('ct.': center, 'cn.': corner) for classification and regression

| CLS | REG | AP | $AP_{50}$ | $AP_{75}$ | $AP_{90}$ |
|-----|-----|-----|-----|-----|-----|
| none | none | 35.6 | 55.5 | 39.0 | 9.3 |
| none | ct. | 36.4 | 54.7 | 38.9 | 10.1 |
| none | cn. | 37.5 | 54.6 | 40.3 | 12.2 |
| ct. | none | 37.3 | 56.6 | 39.9 | 10.5 |
| cn. | none | 36.2 | 55.1 | 38.4 | 9.8 |
| ct. | cn. | **38.5** | **57.0** | **42.3** | **12.6** |

Table 6: Ablation of appearance and geometry terms

| appearance | geometry | AP | $AP_{50}$ | $AP_{75}$ | $AP_{90}$ |
|-----|-----|-----|-----|-----|-----|
| ✗ | ✗ | 35.6 | 55.5 | 39.0 | 9.3 |
| ✓ | ✗ | 37.4 | 56.7 | 41.3 | 10.7 |
| ✗ | ✓ | 37.6 | 55.8 | 41.5 | 12.0 |
| ✓ | ✓ | **38.5** | **57.0** | **42.3** | **12.6** |

Table 7: Compatibility with the non-local module (NL) [33]

| method | AP | $AP_{50}$ | $AP_{75}$ |
|-----|-----|-----|-----|
| RetinaNet | 35.6 | 55.5 | 39.0 |
| RetinaNet + NL | 37.0 | 57.0 | 39.3 |
| RetinaNet + BVR | 38.5 | 57.0 | 42.3 |
| RetinaNet + NL + BVR | **39.4** | **58.2** | **42.5** |

Table 8: Compatibility with the object relation module (ORM) [12]. ResNet-50-FPN is used

| method | AP | $AP_{50}$ | $AP_{75}$ |
|-----|-----|-----|-----|
| faster R-CNN | 37.4 | 58.1 | 40.4 |
| faster R-CNN + ORM | 38.4 | 59.0 | 41.3 |
| faster R-CNN + BVR | 39.3 | 59.5 | 43.1 |
| faster R-CNN + ORM + BVR | **40.4** | **60.6** | **44.0** |

Ablation study of the unit length and the size of the shared location map in Table 4 indicates stable performance. We adopt a unit length of $[4, 8, 16, 32, 64]$ and map size of $400 \times 400$ by default.

**Separate BVR modules for classification and regression** Table 5 ablates the effect of using separate BVR modules for classification and regression, indicating the center representation is a more suitable auxiliary for classification and the corner representation is a more suitable auxiliary for regression.

**Effect of appearance and geometry terms** Table 6 ablates the effect of appearance and geometry terms. Using the two terms together outperforms that using the appearance term alone by 1.1 AP and outperforms that using the geometry term alone by 0.9 AP. In general, the geometry term benefits more at larger IoU criteria, and less at lower IoU criteria.

**Compare with multi-task learning** Only including an auxiliary point head without using it can boost the RetinaNet baseline by 0.8 AP (from 35.6 to 36.4). Noting the BVR brings a 2.9 AP improvement (from 35.6 to 38.5) under the same settings, the major improvements are not due to multi-task learning.

**Complexity analysis** Table 9 shows the flops analysis. The input images are resized to $800 \times 1333$. The proposed BVR module introduces about 3% more parameters (39M vs 38M) and about 10% more computations (266G vs 239G) than the original RetinaNet. We also conduct RetinaNet with heavier head network to have similar parameters and computations as our approach. By adding one more layer, the accuracy slightly drops to 35.2, probably due to the increasing difficulty in optimization. We introduce a GN layer after every head conv layer to alleviate it, and one additional conv layer improves the accuracy by 0.3 AP. These results indicate that the improvements by BVR are mostly not due to more parameters and computation.

The real inference speed of different models using a V100 GPU (fp32 mode is used) are shown in Table 11. By using a ResNet-50 backbone, the BVR module usually takes less than 10% overhead. By using a larger ResNeXt-101-DCN backbone, the BVR module usually takes less than 3% overhead.

Table 9: Complexity analysis

| method | #conv | #ch. | FLOP | param | AP |
|-----|-----|-----|-----|-----|-----|
| RetinaNet | 4 | 256 | 239G | 38M | 35.6 |
| RetinaNet (deep) | 5 | 256 | 265G | 39M | 35.2 |
| RetinaNet (wide) | 4 | 288 | 267G | 39M | 35.6 |
| RetinaNet+BVR | 4 | 256 | 266G | 39M | **38.5** |
| RetinaNet+GN | 4 | 256 | 239G | 38M | 36.5 |
| RetinaNet (deep)+GN | 5 | 256 | 265G | 39M | 36.8 |
| RetinaNet (wide)+GN | 4 | 288 | 267G | 39M | 36.5 |
| RetinaNet+GN+BVR | 4 | 256 | 266G | 39M | **39.2** |

Table 10: BVR for four representative detectors using a ResNeXt-64x4d-101-DCN backbone

| method | AP | $AP_{50}$ | $AP_{75}$ |
|-----|-----|-----|-----|
| RetinaNet | 42.9 | 63.4 | 46.9 |
| RetinaNet + BVR | 44.7 (+1.8) | 64.9 | 49.0 |
| faster R-CNN | 45.0 | 66.2 | 48.8 |
| faster R-CNN + BVR | 46.5 (+1.5) | 67.4 | 50.5 |
| FCOS | 46.1 | 65.0 | 49.6 |
| FCOS + BVR | 47.6 (+1.5) | 66.2 | 51.4 |
| ATSS | 48.3 | 67.1 | 52.6 |
| ATSS + BVR | 50.3 (+2.0) | 69.0 | 55.0 |

Table 11: Time cost of the BVR module.

| method | backbone | FPS | FPS (+BVR) |
|---|---|---|---|
| Faster R-CNN | ResNet-50/ResNeXt-101-DCN | 21.3/7.5 | 19.5/7.3 |
| RetinaNet | ResNet-50/ResNeXt-101-DCN | 18.9/7.0 | 17.4/6.8 |
| FCOS | ResNet-50/ResNeXt-101-DCN | 22.7/7.4 | 20.7/7.2 |
| ATSS | ResNet-50/ResNeXt-101-DCN | 19.6/7.1 | 17.9/6.9 |

Table 12: Results on MS COCO `test-dev` set, '∗' denotes the multi-scale testing

| method | backbone | AP | $AP_{50}$ | $AP_{75}$ | $AP_S$ | $AP_M$ | $AP_L$ |
|---|---|---|---|---|---|---|---|
| DCN v2* [40] | ResNet-101-DCN | 46.0 | 67.9 | 50.8 | 27.8 | 49.1 | 59.5 |
| SNIPER* [27] | ResNet-101 | 46.5 | 67.5 | 52.2 | 30.0 | 49.4 | 58.4 |
| RepPoints* [35] | ResNet-101-DCN | 46.5 | 67.4 | 50.9 | 30.3 | 49.7 | 57.1 |
| MAL* [13] | ResNeXt-101 | 47.0 | 66.1 | 51.2 | 30.2 | 50.1 | 58.9 |
| CentripetalNet* [6] | Hourglass-104 | 48.0 | 65.1 | 51.8 | 29.0 | 50.4 | 59.9 |
| ATSS* [37] | ResNeXt-64x4d-101-DCN | 50.7 | 68.9 | 56.3 | 33.2 | 52.9 | 62.4 |
| TSD* [28] | SENet154-DCN | 51.2 | 71.9 | 56.0 | 33.8 | 54.8 | 64.2 |
| RelationNet++ (our) | ResNeXt-64x4d-101-DCN | 50.3 | 69.0 | 55.0 | 32.8 | 55.0 | **65.8** |
| RelationNet++ (our)* | ResNeXt-64x4d-101-DCN | **52.7** | **70.4** | **58.3** | **35.8** | **55.3** | 64.7 |

## 4.2 BVR is Complementary to Other Attention Mechanisms

The BVR module acts differently compared to the pioneer works of the non-local module [33] and the relation module [12] which also model dependencies between representations. While the BVR module models relationships between different kinds of representations, the latter modules model relationships within the same kinds of representations (pixels [33] and proposal boxes [12]). To compare with the object relation module (ORM) [12], we first apply BVR to enhance RoIAlign features with corner/center representations, the process of which is same as Figure 4a. Then the enhanced features are utilized to perform object relation between proposals. Different from [12], keys are sampled to make the module more efficient. Table 8 shows that the BVR module and the relation module are mostly complementary. On the basis of faster R-CNN baseline, ORM can obtain $+1.0$ AP improvement, while our BVR improves AP by $1.9$. Applying our BVR on the basis of the ORM continually improves AP by $2.0$. Table 7 and 8 show that the BVR modules is mostly complementary with non-local and object relation module.

## 4.3 Generally Applicable to Representative Detectors

We apply the proposed BVR to four representative frameworks, i.e., RetinaNet [18], Faster R-CNN [24, 17], FCOS [29] and ATSS [37], as shown in Table 10. The ResNeXt-64x4d-101-DCN backbone, multi-scale and longer training (20 epochs) are adopted to test whether our approach effects on strong baselines. The BVR module improve these strong detectors by $1.5 \sim 2.0$ AP.

## 4.4 Comparison with State-of-the-Arts

We build our detector by applying the BVR module on a strong detector of ATSS, which achieves $50.7$ AP on COCO `test-dev` using multi-scale testing based on the ResNeXt-64x4d-101-DCN backbone. Our approach improves it by $2.0$ AP, reaching $52.7$ AP. Table 12 shows the comparison with state-of-the-arts methods.

## 5 Conclusion

In this paper, we present a new module, BVR, which bridge various other visual representations by an attention mechanism like that in Transformer [31] to enhance the main representations in a detector. The BVR module can be applied plug-in for an existing detector, and proves broad effectiveness for prevalent object detection frameworks, i.e. RetinaNet, faster R-CNN, FCOS and ATSS, where about $1.5 \sim 3.0$ AP improvements are achieved. We reach $52.7$ AP on COCO test-dev by improving a strong ATSS detector. The resulting network is named RelationNet++, which advances the relation modeling in [12] from bbox-to-bbox to across heterogeneous object/part representations.

## Broader Impact

This work aims for better object detection algorithms. Any object oriented visual applications may benefit from this work, as object detection is usually an indispensable component for them. There may be unpredictable failures, similar as most other detectors. The consequences of failures by this algorithm are determined on the down-stream applications, and please do not use it for scenarios where failures will lead to serious consequences. The method is data driven, and the performance may be affected by the biases in the data. So please also be careful about the data collection process when using it.

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
