[Reviews · NeurIPS 2020]

Review 1

Summary and Contributions: This paper proposes a method to use different object representations (i.e. anchor/proposal, center point, corner points) simultaneously in a single object detection framework. It combines the strengths of each representation. Similar to how features interact with each other in an attention module (such as in Transformers), different representations enforce each other, thereby improving the object detector accuracy. The method is applied to many different detectors (RetinaNet, FCOS, Faster RCNN, ATSS). Let's take RetinaNet as an example. It uses the anchor representation by default. The proposed method adds a lightweight network to RetinaNet, which predicts center and corner points. An anchor of interest is taken as the query and estimated (center and corner) points in its local neighbor are taken as "keys". Through the usual attention formulation, the features of the query is enhanced using the information from the keys. The method is widely applicable and improves all baselines. By improving ATSS, the method achieves 52.7 AP on COCO test-dev.

Strengths: + Very well-written paper. + Proposed method significantly improves a variety of strong baselines. It is widely applicable. + Proposed method achieves SOTA performance on COCO test-dev by improving ATSS.

Weaknesses: -

Correctness: Claims made in the abstract and introduction are backed by experiments. Method seems to be correct. Empirical methodology is correct.

Clarity: Yes, very well. A grammatical error at line 173: "is consist of"

Relation to Prior Work: To the best of my knowledge, all related work is present and they are sufficiently discussed.

Reproducibility: Yes

Additional Feedback: From the authors' response to the "4. ML Reproducibility -- Code" part of the "Submission Questions", I get the impression that authors provide code, however, it is not provided in the supplementary material. Below I provide some comments/suggestions: * The "strength" of each representation type is mentioned without any data (lines 30-31). Would be great to base these on data. * You say that "the method works in-place". Different people might interpret this differently. Please be more specific and describe what you mean. * Why do you say "part corners" to corners? Aren't they box corners? They are not corners of object parts. * I find the term "evolution flow" a bit exaggerated. We are talking about only one-stage or at most two-stages. Evolution? * Section 3.1 describes Eq1 and Eq2 only technically and symbolically. Although these equations come from the well-known attention module, an intuitive description of what is going on would be highly appreciated by the reader. * This method could be even more interesting if it can efficiently handle long-range interactions. Currently, at least for the RetinaNet, it only uses "keys" in a local neighborhood. The authors might want to check a related, very recent paper: HoughNet, ECCV 2020.


Review 2

Summary and Contributions: This paper analyzes three representations of objects: anchor/proposal rectangle boxes, center points, and corner points. The authors claim that different representations have different benefits, and exploit self-attention methods to integrate them together. The method, named bridging visual representations (BVR), has broad effectiveness in current detection frameworks, include RetinaNet, Faster R-CNN, FCOS and ATSS.

Strengths: 1. This paper is well written and easy to follow. 2. The experiments are thorough and validate the effectiveness of the proposed method. 3. The idea in this paper is good that integrate different representations of objects together to boost the performance. The idea is simple and effective. Overall I think this is a good paper. It has reasonable motivations, that is, current object detection systems are doing different representations for objects, this paper want to integrate them together. The methodology proposed in this paper tackle the problems mentioned by the authors. The experiments part is thorough and analyze the details for different aspects. The authors have done some experiments that draw my concerns. For example, they analyze the relations between the proposed module and non-local blocks, relation networks, which is also based on self-attention blocks. The authors analyze the computational cost for their method that also darw my concerns. I think this paper does a good job in academic writing and is self-contained. I recommend to accept it in NeurIPS.

Weaknesses: As I mentioned in strengthens, I think this paper did a good job in academic writing and propose thorough experiments. Most of my concerns during reading the main paper are drawed in their experimental parts. However, the idea does not superise me. Self-attention block is well used in vision tasks. This paper finds a great application for self-attention block and tackle the proposed problem. The authors use some methods to "hard" integrate different representations. There is still master representations to represent the objects. It would be great if we can truly do something that can integrate the representations together in following works.

Correctness: Yes.

Clarity: Yes. The paper is well written.

Relation to Prior Work: Yes.

Reproducibility: Yes

Additional Feedback: Comments after rebuttal: The rebuttal address most concerns of the reviewers. I keep my reviews as accept.


Review 3

Summary and Contributions: In this work, the authors analyzed the representation formats of current object detectors and proposed an attention-based manner to combine the advantages of different representations. The proposed method is compared with various state-of-the-art methods and is shown to be better.

Strengths: 1. The authors concluded the advantages and disadvantages of different object representations. 2. To combine the strengths of different object representations, the authors developed an attention-based module to bridge them. 3. The author optimized the computation and memory complexities of the proposed attention module. 4. The proposed method achieved good performance.

Weaknesses: 1. Some implemented details are not clear. a) In the point head network, are the predictions (including top-k strategies) of top-left points and bottom-right points individual? b) How do the features of two corner points enhance the single regression feature? It seems that there is no detailed explanation about them in the paper, and it would be better if there are figures to illustrate them. 2. The proposed method has an extra point head and two pixel-wise attention layers which are time-consuming. a) What is the input size during the FLOPS calculation? b) What is the time cost of BVR? Can you provide a detailed time cost analysis about BVR?

Correctness: The method is clearly described except for some implemented details and the empirical methodology is correct.

Clarity: Most parts of the paper are well written.

Relation to Prior Work: The authors have discussed the differences from previous works.

Reproducibility: No

Additional Feedback:


Review 4

Summary and Contributions: This paper works on enhancing existing object detectors by adding a center/ corner point branch as an auxiliary task. This additional point branch provides an attention mask for both the existing classification and regression branches. Experiments show the proposed method brings ~2mAP improvements on COCO dataset under different detectors with ~10% more computation. The best performance achieved 52.7 mAP.

Strengths: + The experimental results are strong (~2mAP improvements on different detectors), especially the best performance of 52.7 mAP. + The authors have experimented under different detectors: both one-stage detectors and Faster RCNN, shows the proposed contribution is general.

Weaknesses: - While the overall performance is strong, the reviewer is not excited about the technical novelty. The good performance feels like from putting existing output modalities together. Training cornernet/ centernet in an FPN structure is new, but this part is not well explained in the paper: what is the training loss for the point head? Is that the CornerNet-style focal loss or standard cross-entropy? How to assign different points to different FPN levels? - All ablation studies are done on the poor RetinaNet, where the numbers are not exciting at all. It will be much better if the ablation experiments are on the strongest baseline. The author claimed RetinaNet and ATSS/ FCOS are the same (L217), however it is unclear to the reviewer how the centerness branch in ATSS and FCOS work. Does the centerness branch play a similar role as the center point head? - Here is one interesting baseline: can we only add the point head as an auxiliary task, but not apply it as attention? This will help ablate if the improvement is from the auxiliary task or from the attention. - The paper writing is unsatisfactory. See the Clarity box. - L274 provides the additional computation in FLOPs, however the runtime will be more straightforward. The authors are suggested to report the runtime for both the baselines and the proposed method on their own machines. - The authors claim "a general viewpoint to understand current object detection frameworks" as their first contribution (L56). Unfortunately, the reviewer didn't find any new insights from section 2. The analysis is more of a standard related work rather than a contribution.

Correctness: The main claims are fair as far as the reviewer can assess. There seem to be an misuse of the big O notation is L195.

Clarity: There are a few unsatisfactory parts in writing: - All figures are without comprehensive captions. E.g., what does fig. 2 want to convey? It seems fig. 2. is very similar to fig. 1. - Figure 3 and figure 4 are unclear. First, it has no captions. It will be much helpful if the authors can illustrate the data shape in and out each block. - The query-keys description in Section 3.1 is not a familiar concept for the general object detection audience. It seems just a softmax attention. - The authors mentioned "corners are more accurate in localization" at least twice (L92, L127). This seems intuitive, but are there any experiments are reference can backup this point? - L119 "FCOS uses center points as its representation". As far as the reviewer understands, FCOS uses all points inside a box. Center sampling is not the core information of FCOS.

Relation to Prior Work: Yes.

Reproducibility: No

Additional Feedback: If there is a neutral borderline I would choose that. The only reason to support accepting this paper is its performance. However the contribution is not clear for me yet (see paper weaknesses), and the presentation needs many improvements. I first leave my rating as 5 and hope the authors can address my concerns in the rebuttal. My complaint on the technical novelty is resolved by the "evidence fir strength of different object representations" section of the rebuttal. The fact that different representations are good at different aspects is very interesting and motivates the paper well. The authors also satisfied my curiosity on the multi-task-only baseline, and promised to fix the writing issue. Based on the rebuttal and the reviews from other reviewers, I gladly raise my rating to 7.

[Author Response · NeurIPS 2020]

We thank all reviewers for their constructive and valuable comments.

**(R1) Typos and codes.** We will fix typos in the revision and make the code publicly available.

**(R1&R4) Evidence for strength of different object representations.** We test the detailed metrics for 4 typical models
of different representations under similar overall AP (through different backbones), i.e., Faster R-CNN (proposal, AP
$= 41.0$, $AP_{50} = \textbf{61.3}$, $AP_{90} = 16.1$, $AP_S = 24.0$, $AP_L = 53.5$), RetinaNet (anchor, AP $= 40.8$, $AP_{50} = 60.5$, $AP_{90} =$
$14.6$, $AP_S = 22.9$, $AP_L = 54.6$), FCOS (center, AP $= 40.9$, $AP_{50} = 60.3$, $AP_{90} = 14.3$, $AP_S = \textbf{24.7}$, $AP_L = 52.3$) and
CornerNet (corner, AP $= 40.4$, $AP_{50} = 56.2$, $AP_{90} = \textbf{23.4}$, $AP_S = 20.2$, $AP_L = \textbf{56.3}$). We can see that the bounding box
representation (Faster R-CNN) is more friendly for classification (highest $AP_{50} = 61.3$); center representation (FCOS)
is more friendly to small objects (highest $AP_S = 24.7$), and corner representation (CornerNet) is more accurate for
larger objects and finer localization (highest $AP_{90} = 23.4$ and highest $AP_L = 56.3$).

**(R1) Explanation of "works in-place", "part corners" and "evolution flow".** The proposed module can be directly
plugged into the existing detectors mentioned in the paper. We agree "corners" is a better description. We will also use
"representation flow" instead of "evolution flow" as suggested. These terms will be modified in the revised version.

**(R1&R4) Intuitiveness of Section 3.1.** We will rewrite it to be more intuitive and more friendly to detection audience.

**(R1) Long-range interaction.** We select the top-$k$ key features in the entire feature map, and thus the keys are not
limited in the local range. We will discuss HoughNet in our revision.

**(R2)** Thanks for recognizing our "reasonable motivation" and "simple and effective" method. Also thanks for the great
suggestion of "truly" integrating representations without a "master". We will keep thinking in this direction.

**(R3) Implementation details.** The prediction of left-top and right-bottom points are individual. We merge the enhanced
feature maps of left-top and right-bottom corner by addition. We will make it clearer in revision.

**(R3&R4) Real time cost of the BVR.** Table 9 uses an input size of $800 \times 1333$ to count the FLOPs. The real inference
speed of different models using a V100 GPU (fp32 mode is used) are shown in Table 1. By using a ResNet-50 backbone,
the BVR module usually takes less than $10\%$ overhead. By using a larger ResNeXt-101-DCN backbone, the BVR
module usually takes less than $3\%$ overhead.

Table 1: Time cost of the BVR module. R-50 and RX-101-D mean ResNet-50 and ResNeXt-101-DCN, respectively.

| Method | Backbone | FPS | FPS (+BVR) | Method | Backbone | FPS | FPS (+BVR) |
|---|---|---|---|---|---|---|---|
| Faster R-CNN | R-50/RX-101-D | 21.3/7.5 | 19.5/7.3 | FCOS | R-50/RX-101-D | 22.7/7.4 | 20.7/7.2 |
| RetinaNet | R-50/RX-101-D | 18.9/7.0 | 17.4/6.8 | ATSS | R-50/RX-101-D | 19.6/7.1 | 17.9/6.9 |

**(R4) Good performance feels like from putting existing output modalities together; not excited.** Actually, this is
exactly our goal: while "current object detection systems are doing different representations for objects, this paper wants
to integrate them together" (by R2). To achieve this goal, we propose an attention module to bridge these heterogeneous
representations. We also propose *novel* techniques of *key sampling* and *shared location embedding* to make the module
effective and efficient. Our approach is "simple, effective" (by R2) and general for many different detectors including
RetinaNet, FCOS, Faster R-CNN and ATSS (by R1). Although we respect the personal view of "not excited", we would
greatly appreciate if the reviewer could also think about "the reasonable motivation" (by R2), "the simple and effective
nature" (by R2) and the broad effectiveness (by R1, R2) of this submission.

**(R4) Novel CornerNet/CenterNet in an FPN but without details.** Thanks for recognizing this novelty, although
there are other contributions we value more greatly as described in the response to the last question. The details are in
Lines 179-184: we use a CornerNet-style focal loss and assign all ground-truth center/corner points to all FPN levels.

**(R4) "ablation experiments on the strongest baseline".** Thanks for the suggestion. Noticing BVR achieves 2.0 AP
gain on our strongest baseline detector of ATSS (see Table 10), we think whether the proposed technique works well on
stronger baselines is well answered. Ablation on the strongest baseline is good, but a more common practice for most
papers is to ablate on a reasonably well baseline due to resource limit.

**(R4) The centerness branch and center point head.** FCOS and ATSS include a classification and a regression branch
on the "centerness" points. When applying BVR, we use an independent center point head and corner point head. The
center point head plays a different role than the centerness branch: center point is an *auxiliary* representation referring
to the box center and aims to strenghthen the *master* features, while centerness is a *master* representation referring
to box (center) area and aims for classification and regression. Nevertheless, they could share a same branch, but to
strengthen the commonality of the BVR module, we use a same and independent center/corner point heads for all
baseline detectors no matter what the master branches are.

**(R4) Gains by multi-task learning.** Only including an auxiliary point head without using it can boost the RetinaNet
baseline by 0.8 AP (from 35.6 to 36.4). Noting the BVR brings a 2.9 AP improvement (from 35.6 to 38.5) under the
same settings, the major improvements are not due to multi-task learning. We will add this ablation in our revision.

**(R4) A general viewpoint as a contribution.** We shall remove it as a contribution in our revision.

**(R4) Figures.** Figure 1b illustrates typical object representations, while Figure 2 focus on how representation flows
throughout the detection pipeline. We will follow the reviewer's suggestions to provide detailed captions for each figure.

**(R4) Representation in FCOS.** The updated version of FCOS "samples the central portion of ground-truth boxes as
positive" (see "ctr. sampling" in Table 3 of FCOS), and we use an implementation of this version in our experiments.
We thus categorize it as a center based method. We will make it clearer in our revision.

[Meta-Review · NeurIPS 2020]

Good work on analyzing pros and cons of various object representations, as well as a neat way to combine them into a single framework that gives good gains on the COCO benchmark. The proposed solution of using a self-attention module to bridge the representations is both original, simple and widely-applicable. I think the method and the work reveal intriguing differences between the various representations and this will be useful to the community. The authors should adapt the camera ready in accordance to the post-rebuttal comments from the reviewers (esp. as it concerns more fine-grained statements about the contributions of this work).